# Learning Spherical Convolution
# for Fast Features from 360° Imagery

**Yu-Chuan Su**      **Kristen Grauman**
The University of Texas at Austin

## Abstract

While 360° cameras offer tremendous new possibilities in vision, graphics, and augmented reality, the spherical images they produce make core feature extraction non-trivial. Convolutional neural networks (CNNs) trained on images from perspective cameras yield "flat" filters, yet 360° images cannot be projected to a single plane without significant distortion. A naive solution that repeatedly projects the viewing sphere to all tangent planes is accurate, but much too computationally intensive for real problems. We propose to *learn* a spherical convolutional network that translates a planar CNN to process 360° imagery directly in its equirectangular projection. Our approach learns to reproduce the flat filter outputs on 360° data, sensitive to the varying distortion effects across the viewing sphere. The key benefits are 1) efficient feature extraction for 360° images and video, and 2) the ability to leverage powerful pre-trained networks researchers have carefully honed (together with massive labeled image training sets) for perspective images. We validate our approach compared to several alternative methods in terms of both raw CNN output accuracy as well as applying a state-of-the-art "flat" object detector to 360° data. Our method yields the most accurate results while saving orders of magnitude in computation versus the existing exact reprojection solution.

## 1 Introduction

Unlike a traditional perspective camera, which samples a limited field of view of the 3D scene projected onto a 2D plane, a 360° camera captures the entire viewing sphere surrounding its optical center, providing a complete picture of the visual world—an omnidirectional field of view. As such, viewing 360° imagery provides a more immersive experience of the visual content compared to traditional media.

360° cameras are gaining popularity as part of the rising trend of virtual reality (VR) and augmented reality (AR) technologies, and will also be increasingly influential for wearable cameras, autonomous mobile robots, and video-based security applications. Consumer level 360° cameras are now common on the market, and media sharing sites such as Facebook and YouTube have enabled support for 360° content. For consumers and artists, 360° cameras free the photographer from making real-time composition decisions. For VR/AR, 360° data is essential to content creation. As a result of this great potential, computer vision problems targeting 360° content are capturing the attention of both the research community and application developer.

Immediately, this raises the question: how to compute features from 360° images and videos? Arguably the most powerful tools in computer vision today are convolutional neural networks (CNN). CNNs are responsible for state-of-the-art results across a wide range of vision problems, including image recognition [17, 42], object detection [12, 30], image and video segmentation [16, 21, 28], and action detection [10, 32]. Furthermore, significant research effort over the last five years (and really decades [27]) has led to well-honed CNN architectures that, when trained with massive labeled image datasets [8], produce "pre-trained" networks broadly useful as feature extractors for new problems.

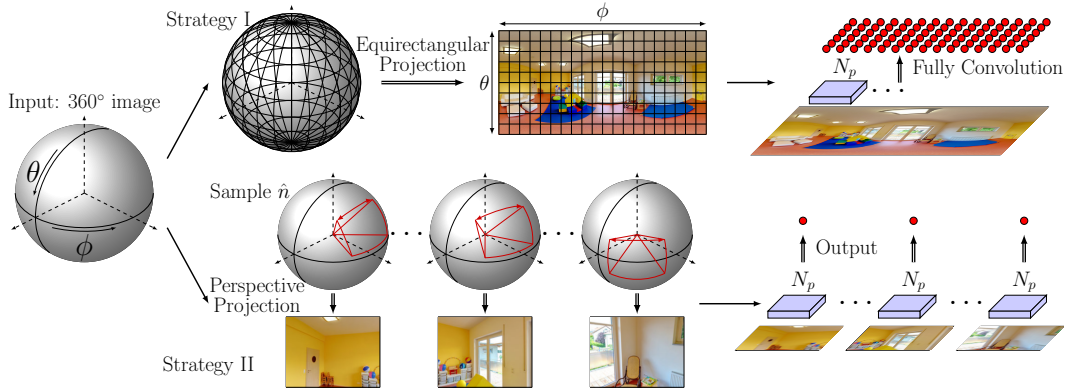

**Figure 1:** Two existing strategies for applying CNNs to 360° images. **Top:** The first strategy unwraps the 360° input into a single planar image using a global projection (most commonly equirectangular projection), then applies the CNN on the distorted planar image. **Bottom:** The second strategy samples multiple tangent planar projections to obtain multiple perspective images, to which the CNN is applied independently to obtain local results for the original 360° image. Strategy I is fast but inaccurate; Strategy II is accurate but slow. The proposed approach learns to replicate flat filters on spherical imagery, offering both speed and accuracy.

Indeed such networks are widely adopted as off-the-shelf feature extractors for other algorithms and applications (c.f., VGG [33], ResNet [17], and AlexNet [25] for images; C3D [36] for video).

However, thus far, powerful CNN features are awkward if not off limits in practice for 360° imagery. The problem is that the underlying projection models of current CNNs and 360° data are different. Both the existing CNN filters *and* the expensive training data that produced them are "flat", i.e., the product of perspective projection to a plane. In contrast, a 360° image is projected onto the unit sphere surrounding the camera's optical center.

To address this discrepancy, there are two common, though flawed, approaches. In the first, the spherical image is projected to a planar one,[1] then the CNN is applied to the resulting 2D image [19,26] (see Fig. 1, top). However, any sphere-to-plane projection introduces distortion, making the resulting convolutions inaccurate. In the second existing strategy, the 360° image is repeatedly projected to tangent planes around the sphere, each of which is then fed to the CNN [34, 35, 38, 41] (Fig. 1, bottom). In the extreme of sampling every tangent plane, this solution is exact and therefore accurate. However, it suffers from very high computational cost. Not only does it incur the cost of rendering each planar view, but also it prevents amortization of convolutions: the intermediate representation cannot be shared across perspective images because they are projected to different planes.

We propose a learning-based solution that, unlike the existing strategies, sacrifices neither accuracy nor efficiency. The main idea is to learn a CNN that processes a 360° image in its equirectangular projection (fast) but mimics the "flat" filter responses that an existing network would produce on all tangent plane projections for the original spherical image (accurate). Because convolutions are indexed by spherical coordinates, we refer to our method as *spherical convolution* (SPHCONV). We develop a systematic procedure to adjust the network structure in order to account for distortions. Furthermore, we propose a kernel-wise pre-training procedure which significantly accelerates the training process.

In addition to providing fast general feature extraction for 360° imagery, our approach provides a bridge from 360° content to existing heavily supervised datasets dedicated to perspective images. In particular, training requires no new annotations—only the target CNN model (e.g., VGG [33] pre-trained on millions of labeled images) and an arbitrary collection of unlabeled 360° images.

We evaluate SPHCONV on the Pano2Vid [35] and PASCAL VOC [9] datasets, both for raw convolution accuracy as well as impact on an object detection task. We show that it produces more precise outputs than baseline methods requiring similar computational cost, and similarly precise outputs as the exact solution while using orders of magnitude less computation. Furthermore, we demonstrate that SPHCONV can successfully replicate the widely used Faster-RCNN [30] detector on 360° data when training with only 1,000 unlabeled 360° images containing unrelated objects. For a similar cost as the baselines, SPHCONV generates better object proposals and recognition rates.

## 2 Related Work

**360° vision**   Vision for 360° data is quickly gaining interest in recent years. The SUN360 project samples multiple perspective images to perform scene viewpoint recognition [38]. PanoContext [41] parses 360° images using 3D bounding boxes, applying algorithms like line detection on perspective images then backprojecting results to the sphere. Motivated by the limitations of existing interfaces for viewing 360° video, several methods study how to automate field-of-view (FOV) control for display [19, 26, 34, 35], adopting one of the two existing strategies for convolutions (Fig. 1). In these methods, a noted bottleneck is feature extraction cost, which is hampered by repeated sampling of perspective images/frames, e.g., to represent the space-time "glimpses" of [34, 35]. This is exactly where our work can have positive impact. Prior work studies the impact of panoramic or wide angle images on hand-crafted features like SIFT [11, 14, 15]. While not applicable to CNNs, such work supports the need for features specific to 360° imagery, and thus motivates SPHCONV.

**Knowledge distillation**   Our approach relates to knowledge distillation [3, 5, 13, 18, 29, 31, 37], though we explore it in an entirely novel setting. Distillation aims to learn a new model given existing model(s). Rather than optimize an objective function on annotated data, it learns the new model that can reproduce the behavior of the existing model, by minimizing the difference between their outputs. Most prior work explores distillation for model compression [3, 5, 18, 31]. For example, a deep network can be distilled into a shallower [3] or thinner [31] one, or an ensemble can be compressed to a single model [18]. Rather than compress a model in the same domain, our goal is to learn *across* domains, namely to link networks on images with different projection models. Limited work considers distillation for transfer [13, 29]. In particular, unlabeled target-source paired data can help learn a CNN for a domain lacking labeled instances (e.g., RGB vs. depth images) [13], and multi-task policies can be learned to simulate action value distributions of expert policies [29]. Our problem can also be seen as a form of transfer, though for a novel task motivated strongly by image processing complexity as well as supervision costs. Different from any of the above, we show how to adapt the network structure to account for geometric transformations caused by different projections. Also, whereas most prior work uses only the final output for supervision, we use the intermediate representation of the target network as both input and target output to enable kernel-wise pre-training.

**Spherical image projection**   Projecting a spherical image into a planar image is a long studied problem. There exists a large number of projection approaches (e.g., equirectangular, Mercator, etc.) [4]. None is perfect; every projection must introduce some form of distortion. The properties of different projections are analyzed in the context of displaying panoramic images [40]. In this work, we unwrap the spherical images using equirectangular projection because 1) this is a very common format used by camera vendors and researchers [1, 35, 38], and 2) it is equidistant along each row and column so the convolution kernel does not depend on the azimuthal angle. Our method in principle could be applied to other projections; their effect on the convolution operation remains to be studied.

**CNNs with geometric transformations**   There is an increasing interest in generalizing convolution in CNNs to handle geometric transformations or deformations. Spatial transformer networks (STNs) [20] represent a geometric transformation as a sampling layer and predict the transformation parameters based on input data. STNs assume the transformation is invertible such that the subsequent convolution can be performed on data without the transformation. This is not possible in spherical images because it requires a projection that introduces no distortion. Active convolution [22] learns the kernel shape together with the weights for a more general receptive field, and deformable convolution [7] goes one step further by predicting the receptive field location. These methods are too restrictive for spherical convolution, because they require a fixed kernel size and weight. In contrast, our method adapts the kernel size and weight based on the transformation to achieve better accuracy. Furthermore, our method exploits problem-specific geometric information for efficient training and testing. Some recent work studies convolution on a sphere [6, 24] using spectral analysis, but those methods require manually annotated spherical images as training data, whereas our method can exploit existing models trained on perspective images as supervision. Also, it is unclear whether CNNs in the spectral domain can reach the same accuracy and efficiency as CNNs on a regular grid.

## 3 Approach

We describe how to learn spherical convolutions in equirectangular projection given a target network trained on perspective images. We define the objective in Sec. 3.1. Next, we introduce how to adapt the structure from the target network in Sec. 3.2. Finally, Sec. 3.3 presents our training process.

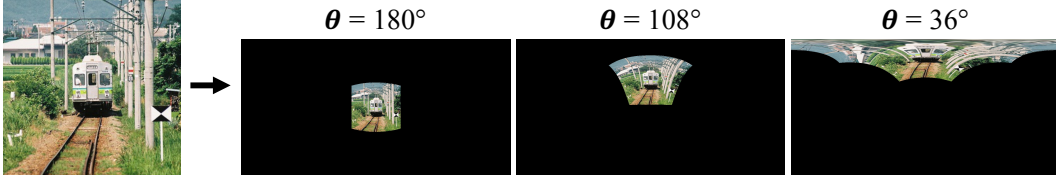

$\theta = 180°$     $\theta = 108°$     $\theta = 36°$

**Figure 2:** Inverse perspective projections $\mathcal{P}^{-1}$ to equirectangular projections at different polar angles $\theta$. The same square image will distort to different sizes and shapes depending on $\theta$. Because equirectangular projection unwraps the 180° longitude, a line will be split into two if it passes through the 180° longitude, which causes the double curve in $\theta = 36°$.

## 3.1 Problem Definition

Let $I_s$ be the input spherical image defined on spherical coordinates $(\theta, \phi)$, and let $I_e \in \mathbb{I}^{W_e \times H_e \times 3}$ be the corresponding flat RGB image in equirectangular projection. $I_e$ is defined by pixels on the image coordinates $(x, y) \in D^e$, where each $(x, y)$ is linearly mapped to a unique $(\theta, \phi)$. We define the perspective projection operator $\mathcal{P}$ which projects an $\alpha$-degree field of view (FOV) from $I_s$ to $W$ pixels on the the tangent plane $\hat{n} = (\theta, \phi)$. That is, $\mathcal{P}(I_s, \hat{n}) = I_p \in \mathbb{I}^{W \times W \times 3}$. The projection operator is characterized by the pixel size $\Delta_p \theta = \alpha/W$ in $I_p$, and $I_p$ denotes the resulting perspective image. Note that we assume $\Delta\theta = \Delta\phi$ following common digital imagery.

Given a target network[2] $N_p$ trained on perspective images $I_p$ with receptive field (Rf) $R \times R$, we define the output on spherical image $I_s$ at $\hat{n} = (\theta, \phi)$ as

$$N_p(I_s)[\theta, \phi] = N_p(\mathcal{P}(I_s, (\theta, \phi))), \tag{1}$$

where w.l.o.g. we assume $W = R$ for simplicity. Our goal is to learn a spherical convolution network $N_e$ that takes an equirectangular map $I_e$ as input and, for every image position $(x, y)$, produces as output the results of applying the perspective projection network to the corresponding tangent plane for spherical image $I_s$:

$$N_e(I_e)[x, y] \approx N_p(I_s)[\theta, \phi], \quad \forall (x, y) \in D^e, \quad (\theta, \phi) = \left(\frac{180° \times y}{H_e}, \frac{360° \times x}{W_e}\right). \tag{2}$$

This can be seen as a domain adaptation problem where we want to transfer the model from the domain of $I_p$ to that of $I_e$. However, unlike typical domain adaptation problems, the difference between $I_p$ and $I_e$ is characterized by a geometric projection transformation rather than a shift in data distribution. Note that the training data to learn $N_e$ requires no manual annotations: it consists of arbitrary 360° images coupled with the "true" $N_p$ outputs computed by exhaustive planar reprojections, i.e., evaluating the rhs of Eq. 1 for every $(\theta, \phi)$. Furthermore, at *test* time, only a single equirectangular projection of the entire 360° input will be computed using $N_e$ to obtain the dense (inferred) $N_p$ outputs, which would otherwise require multiple projections and evaluations of $N_p$.

## 3.2 Network Structure

The main challenge for transferring $N_p$ to $N_e$ is the distortion introduced by equirectangular projection. The distortion is location dependent—a $k \times k$ square in perspective projection will not be a square in the equirectangular projection, and its shape and size will depend on the polar angle $\theta$. See Fig. 2. The convolution kernel should transform accordingly. Our approach 1) adjusts the shape of the convolution kernel to account for the distortion, in particular the content expansion, and 2) reduces the number of max-pooling layers to match the pixel sizes in $N_e$ and $N_p$, as we detail next.

We adapt the architecture of $N_e$ from $N_p$ using the following heuristic. The goal is to ensure each kernel receives enough information from the input in order to compute the target output. First, we untie the weight of convolution kernels at different $\theta$ by learning one kernel $K_e^y$ for each output row $y$. Next, we adjust the shape of $K_e^y$ such that it covers the Rf of the original kernel. We consider $K_e^y \in N_e$ to cover $K_p \in N_p$ if more than 95% of pixels in the Rf of $K_p$ are also in the Rf of $K_e$ in $I_e$. The Rf of $K_p$ in $I_e$ is obtained by backprojecting the $R \times R$ grid to $\hat{n} = (\theta, 0)$ using $\mathcal{P}^{-1}$, where the center of the grid aligns on $\hat{n}$. $K_e$ should be large enough to cover $K_p$, but it should also be as small as possible to avoid overfitting. Therefore, we optimize the shape of $K_e^{l,y}$ for layer $l$ as follows. The shape of $K_e^{l,y}$ is initialized as $3 \times 3$. We first adjust the height $k_h$ and increase $k_h$ by 2

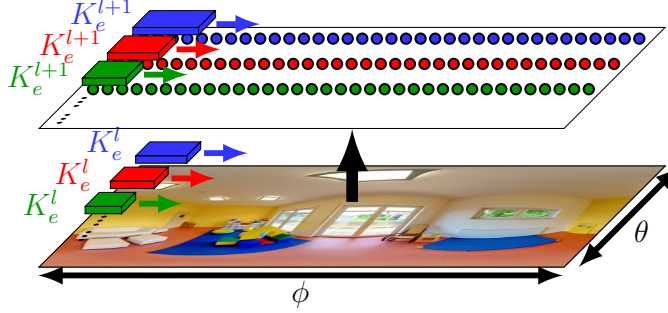

**Figure 3:** Spherical convolution. The kernel weight in spherical convolution is tied only along each row of the equirectangular image (i.e., $\phi$), and each kernel convolves along the row to generate 1D output. Note that the kernel size differs at different rows and layers, and it expands near the top and bottom of the image.

until the height of the Rf is larger than that of $K_p$ in $I_e$. We then adjust the width $k_w$ similar to $k_h$. Furthermore, we restrict the kernel size $k_h \times k_w$ to be smaller than an upper bound $U_k$. See Fig. 4. Because the Rf of $K_e^l$ depends on $K_e^{l-1}$, we search for the kernel size starting from the bottom layer.

It is important to relax the kernel from being square to being rectangular, because equirectangular projection will expand content horizontally near the poles of the sphere (see Fig. 2). If we restrict the kernel to be square, the Rf of $K_e$ can easily be taller but narrower than that of $K_p$ which leads to overfitting. It is also important to restrict the kernel size, otherwise the kernel can grow wide rapidly near the poles and eventually cover the entire row. Although cutting off the kernel size may lead to information loss, the loss is not significant in practice because pixels in equirectangular projection do not distribute on the unit sphere uniformly; they are denser near the pole, and the pixels are by nature redundant in the region where the kernel size expands dramatically.

Besides adjusting the kernel sizes, we also adjust the number of pooling layers to match the pixel size $\Delta\theta$ in $N_e$ and $N_p$. We define $\Delta\theta_e = 180°/H_e$ and restrict $W_e = 2H_e$ to ensure $\Delta\theta_e = \Delta\phi_e$. Because max-pooling introduces shift invariance up to $k_w$ pixels in the image, which corresponds to $k_w \times \Delta\theta$ degrees on the unit sphere, the physical meaning of max-pooling depends on the pixel size. Since the pixel size is usually larger in $I_e$ and max-pooling increases the pixel size by a factor of $k_w$, we remove the pooling layer in $N_e$ if $\Delta\theta_e \geq \Delta\theta_p$.

Fig. 3 illustrates how spherical convolution differs from ordinary CNN. Note that we approximate one layer in $N_p$ by one layer in $N_e$, so the number of layers and output channels in each layer is exactly the same as the target network. However, this does not have to be the case. For example, we could use two or more layers to approximate each layer in $N_p$. Although doing so may improve accuracy, it would also introduce significant overhead, so we stick with the one-to-one mapping.

### 3.3 Training Process

Given the goal in Eq. 2 and the architecture described in Sec. 3.2, we would like to learn the network $N_e$ by minimizing the $L_2$ loss $E[(N_e(I_e) - N_p(I_s))^2]$. However, the network converges slowly, possibly due to the large number of parameters. Instead, we propose a kernel-wise pre-training process that disassembles the network and initially learns each kernel independently.

To perform kernel-wise pre-training, we further require $N_e$ to generate the same intermediate representation as $N_p$ in all layers $l$:

$$N_e^l(I_e)[x, y] \approx N_p^l(I_s)[\theta, \phi] \quad \forall l \in N_e. \tag{3}$$

Given Eq. 3, every layer $l \in N_e$ is independent of each other. In fact, every kernel is independent and can be learned separately. We learn each kernel by taking the "ground truth" value of the previous layer $N_p^{l-1}(I_s)$ as input and minimizing the $L_2$ loss $E[(N_e^l(I_e) - N_p^l(I_s))^2]$, except for the first layer. Note that $N_p^l$ refers to the convolution output of layer $l$ before applying any non-linear operation, e.g. ReLU, max-pooling, etc. It is important to learn the target value before applying ReLU because it provides more information. We combine the non-linear operation with $K_e^{l+1}$ during kernel-wise pre-training, and we use dilated convolution [39] to increase the Rf size instead of performing max-pooling on the input feature map.

For the first convolution layer, we derive the analytic solution directly. The projection operator $\mathcal{P}$ is linear in the pixels in equirectangular projection: $\mathcal{P}(I_s, \hat{n})[x, y] = \sum_{ij} c_{ij} I_e[i, j]$, for coefficients $c_{ij}$

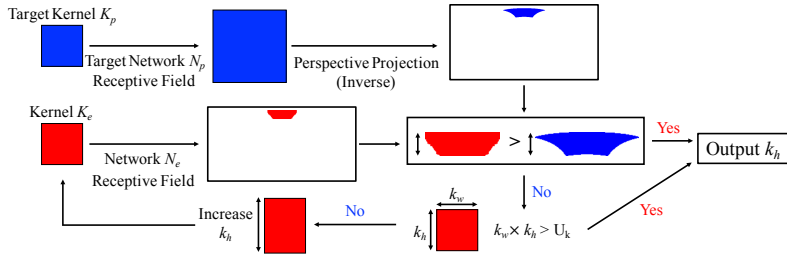

**Figure 4:** Method to select the kernel height $k_h$. We project the receptive field of the target kernel to equirectangular projection $I_e$ and increase $k_h$ until it is taller than the target kernel in $I_e$. The kernel width $k_w$ is determined using the same procedure after $k_h$ is set. We restrict the kernel size $k_w \times k_h$ by an upper bound $U_k$.

from, e.g., bilinear interpolation. Because convolution is a weighted sum of input pixels $K_p * I_p = \sum_{xy} w_{xy} I_p[x, y]$, we can combine the weight $w_{xy}$ and interpolation coefficient $c_{ij}$ as a single convolution operator:

$$K_p^1 * I_s[\theta, \phi] = \sum_{xy} w_{xy} \sum_{ij} c_{ij} I_e[i, j] = \sum_{ij} \left( \sum_{xy} w_{xy} c_{ij} \right) I_e[i, j] = K_e^1 * I_e. \qquad (4)$$

The output value of $N_e^1$ will be exact and requires no learning. Of course, the same is not possible for $l > 1$ because of the non-linear operations between layers.

After kernel-wise pre-training, we can further fine-tune the network *jointly* across layers and kernels by minimizing the $L_2$ loss of the final output. Because the pre-trained kernels cannot fully recover the intermediate representation, fine-tuning can help to adjust the weights to account for residual errors. We ignore the constraint introduced in Eq. 3 when performing fine-tuning. Although Eq. 3 is necessary for kernel-wise pre-training, it restricts the expressive power of $N_e$ and degrades the performance if we only care about the final output. Nevertheless, the weights learned by kernel-wise pre-training are a very good initialization in practice, and we typically only need to fine-tune the network for a few epochs.

One limitation of SPHCONV is that it cannot handle very close objects that span a large FOV. Because the goal of SPHCONV is to reproduce the behavior of models trained on perspective images, the capability and performance of the model is bounded by the target model $N_p$. However, perspective cameras can only capture a small portion of a very close object in the FOV, and very close objects are usually not available in the training data of the target model $N_p$. Therefore, even though 360° images offer a much wider FOV, SPHCONV inherits the limitations of $N_p$, and may not recognize very close large objects. Another limitation of SPHCONV is the resulting model size. Because it unties the kernel weights along $\theta$, the model size grows linearly with the equirectangular image height. The model size can easily grow to tens of gigabytes as the image resolution increases.

## 4   Experiments

To evaluate our approach, we consider both the accuracy of its convolutions as well as its applicability for object detections in 360° data. We use the VGG architecture[3] and the Faster-RCNN [30] model as our target network $N_p$. We learn a network $N_e$ to produce the topmost (conv5_3) convolution output.

**Datasets**   We use two datasets: Pano2Vid for training, and Pano2Vid and PASCAL for testing.

*Pano2Vid:* We sample frames from the 360° videos in the Pano2Vid dataset [35] for both training and testing. The dataset consists of 86 videos crawled from YouTube using four keywords: "Hiking," "Mountain Climbing," "Parade," and "Soccer". We sample frames at 0.05fps to obtain 1,056 frames for training and 168 frames for testing. We use "Mountain Climbing" for testing and others for training, so the training and testing frames are from disjoint videos. See Supp. for sampling process. Because the supervision is on a per pixel basis, this corresponds to $N \times W_e \times H_e \approx 250M$ (non i.i.d.) samples. Note that most object categories targeted by the Faster-RCNN detector do not appear in Pano2Vid, meaning that our experiments test the content-independence of our approach.

*PASCAL VOC:* Because the target model was originally trained and evaluated on PASCAL VOC 2007, we "360-ify" it to evaluate the object detector application. We test with the 4,952 PASCAL images, which contain 12,032 bounding boxes. We transform them to equirectangular images as if they

originated from a 360° camera. In particular, each object bounding box is backprojected to 3 different scales $\{0.5R, 1.0R, 1.5R\}$ and 5 different polar angles $\theta \in \{36°, 72°, 108°, 144°, 180°\}$ on the 360° image sphere using the inverse perspective projection, where $R$ is the resolution of the target network's Rf. Regions outside the bounding box are zero-padded. See Supp. for details. Backprojection allows us to evaluate the performance at different levels of distortion in the equirectangular projection.

**Metrics** We generate the output widely used in the literature (conv5_3) and evaluate it with the following metrics.

*Network output error* measures the difference between $N_e(I_e)$ and $N_p(I_s)$. In particular, we report the root-mean-square error (RMSE) over all pixels and channels. For PASCAL, we measure the error over the Rf of the detector network.

*Detector network performance* measures the performance of the detector network in Faster-RCNN using multi-class classification accuracy. We replace the ROI-pooling in Faster-RCNN by pooling over the bounding box in $I_e$. Note that the bounding box is backprojected to equirectangular projection and is no longer a square region.

*Proposal network performance* evaluates the proposal network in Faster-RCNN using average Intersection-over-Union (IoU). For each bounding box centered at $\hat{n}$, we project the conv5_3 output to the tangent plane $\hat{n}$ using $\mathcal{P}$ and apply the proposal network at the center of the bounding box on the tangent plane. Given the predicted proposals, we compute the IoUs between foreground proposals and the bounding box and take the maximum. The IoU is set to 0 if there is no foreground proposal. Finally, we average the IoU over bounding boxes.

We stress that our goal is not to build a new object detector; rather, we aim to reproduce the behavior of existing 2D models on 360° data with lower computational cost. Thus, the metrics capture how accurately and how quickly we can replicate the *exact* solution.

**Baselines** We compare our method with the following baselines.

- EXACT — Compute the true target value $N_p(I_s)[\theta, \phi]$ for every pixel. This serves as an upper bound in performance and does not consider the computational cost.
- DIRECT — Apply $N_p$ on $I_e$ directly. We replace max-pooling with dilated convolution to produce a full resolution output. This is Strategy I in Fig. 1 and is used in 360° video analysis [19, 26].
- INTERP — Compute $N_p(I_s)[\theta, \phi]$ every $S$-pixels and interpolate the values for the others. We set $S$ such that the computational cost is roughly the same as our SPHCONV. This is a more efficient variant of Strategy II in Fig. 1.
- PERSPECT — Project $I_s$ onto a cube map [2] and then apply $N_p$ on each face of the cube, which is a perspective image with 90° FOV. The result is backprojected to $I_e$ to obtain the feature on $I_e$. We use $W{=}960$ for the cube map resolution so $\Delta\theta$ is roughly the same as $I_p$. This is a second variant of Strategy II in Fig. 1 used in PanoContext [41].

**SPHCONV variants** We evaluate three variants of our approach:

- OPTSPHCONV — To compute the output for each layer $l$, OPTSPHCONV computes the exact output for layer $l{-}1$ using $N_p(I_s)$ then applies spherical convolution for layer $l$. OPTSPHCONV serves as an upper bound for our approach, where it avoids accumulating any error across layers.
- SPHCONV-PRE — Uses the weights from kernel-wise pre-training directly without fine-tuning.
- SPHCONV — The full spherical convolution with joint fine-tuning of all layers.

**Implementation details** We set the resolution of $I_e$ to $640{\times}320$. For the projection operator $\mathcal{P}$, we map $\alpha{=}65.5°$ to $W{=}640$ pixels following SUN360 [38]. The pixel size is therefore $\Delta\theta_e{=}360°/640$ for $I_e$ and $\Delta\theta_p{=}65.5°/640$ for $I_p$. Accordingly, we remove the first three max-pooling layers so $N_e$ has only one max-pooling layer following conv4_3. The kernel size upper bound $U_k{=}7 \times 7$ following the max kernel size in VGG. We insert batch normalization for conv4_1 to conv5_3. See Supp. for details.

### 4.1 Network output accuracy and computational cost

Fig. 5a shows the output error of layers conv3_3 and conv5_3 on the Pano2Vid [35] dataset (see Supp. for similar results on other layers.). The error is normalized by that of the mean predictor. We evaluate the error at 5 polar angles $\theta$ uniformly sampled from the northern hemisphere, since error is roughly symmetric with the equator.

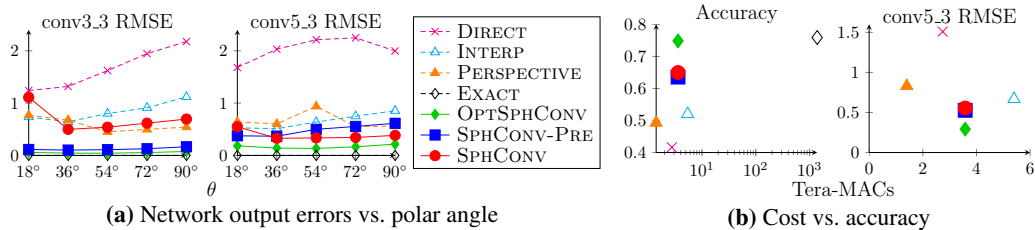

**(a)** Network output errors vs. polar angle

**(b)** Cost vs. accuracy

**Figure 5:** (a) Network output error on Pano2Vid; lower is better. Note the error of EXACT is 0 by definition. Our method's convolutions are much closer to the exact solution than the baselines'. (b) Computational cost vs. accuracy on PASCAL. Our approach yields accuracy closest to the exact solution while requiring orders of magnitude less computation time (left plot). Our cost is similar to the other approximations tested (right plot). Plot titles indicate the y-labels, and error is measured by root-mean-square-error (RMSE).

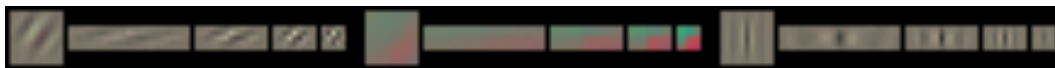

**Figure 6:** Three AlexNet conv1 kernels (left squares) and their corresponding four SPHCONV-PRE kernels at $\theta \in \{9°, 18°, 36°, 72°\}$ (left to right).

First we discuss the three variants of our method. OPTSPHCONV performs the best in all layers and $\theta$, validating our main idea of spherical convolution. It performs particularly well in the lower layers, because the Rf is larger in higher layers and the distortion becomes more significant. Overall, SPHCONV-PRE performs the second best, but as to be expected, the gap with OPTCONV becomes larger in higher layers because of error propagation. SPHCONV outperforms SPHCONV-PRE in conv5_3 at the cost of larger error in lower layers (as seen here for conv3_3). It also has larger error at $\theta=18°$ for two possible reasons. First, the learning curve indicates that the network learns more slowly near the pole, possibly because the Rf is larger and the pixels degenerate. Second, we optimize the joint $L_2$ loss, which may trade the error near the pole with that at the center.

Comparing to the baselines, we see that ours achieves lowest errors. DIRECT performs the worst among all methods, underscoring that convolutions on the flattened sphere—though fast—are inadequate. INTERP performs better than DIRECT, and the error decreases in higher layers. This is because the Rf is larger in the higher layers, so the $S$-pixel shift in $I_e$ causes relatively smaller changes in the Rf and therefore the network output. PERSPECTIVE performs similarly in different layers and outperforms INTERP in lower layers. The error of PERSPECTIVE is particularly large at $\theta=54°$, which is close to the boundary of the perspective image and has larger perspective distortion.

Fig. 5b shows the accuracy vs. cost tradeoff. We measure computational cost by the number of Multiply-Accumulate (MAC) operations. The leftmost plot shows cost on a log scale. Here we see that EXACT—whose outputs we wish to replicate—is about 400 times slower than SPHCONV, and SPHCONV approaches EXACT's detector accuracy much better than all baselines. The second plot shows that SPHCONV is about $34\%$ faster than INTERP (while performing better in all metrics). PERSPECTIVE is the fastest among all methods and is $60\%$ faster than SPHCONV, followed by DIRECT which is $23\%$ faster than SPHCONV. However, both baselines are noticeably inferior in accuracy compared to SPHCONV.

To visualize what our approach has learned, we learn the first layer of the AlexNet [25] model provided by the Caffe package [23] and examine the resulting kernels. Fig. 6 shows the original kernel $K_p$ and the corresponding kernels $K_e$ at different polar angles $\theta$. $K_e$ is usually the re-scaled version of $K_p$, but the weights are often amplified because multiple pixels in $K_p$ fall to the same pixel in $K_e$ like the second example. We also observe situations where the high frequency signal in the kernel is reduced, like the third example, possibly because the kernel is smaller. Note that we learn the first convolution layer for visualization purposes only, since $l = 1$ (only) has an analytic solution (cf. Sec 3.3). See Supp. for the complete set of kernels.

### 4.2 Object detection and proposal accuracy

Having established our approach provides accurate and efficient $N_e$ convolutions, we now examine how important that accuracy is to object detection on 360° inputs. Fig. 7a shows the result of the Faster-RCNN detector network on PASCAL in 360° format. OPTSPHCONV performs almost as well as EXACT. The performance degrades in SPHCONV-PRE because of error accumulation, but it still

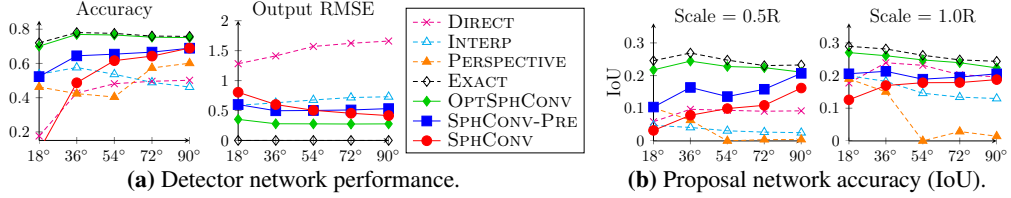

**(a)** Detector network performance.                    **(b)** Proposal network accuracy (IoU).

**Figure 7:** Faster-RCNN object detection accuracy on a 360° version of PASCAL across polar angles $\theta$, for both the (a) detector network and (b) proposal network. $R$ refers to the Rf of $N_p$. Best viewed in color.

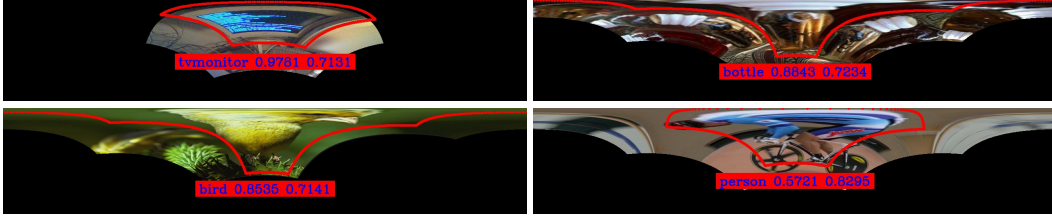

**Figure 8:** Object detection examples on 360° PASCAL test images. Images show the top 40% of equirectangular projection; black regions are undefined pixels. Text gives predicted label, multi-class probability, and IoU, resp. Our method successfully detects objects undergoing severe distortion, some of which are barely recognizable even for a human viewer.

significantly outperforms DIRECT and is better than INTERP and PERSPECTIVE in most regions. Although joint training (SPHCONV) improves the output error near the equator, the error is larger near the pole which degrades the detector performance. Note that the Rf of the detector network spans multiple rows, so the error is the weighted sum of the error at different rows. The result, together with Fig. 5a, suggest that SPHCONV reduces the conv5_3 error in parts of the Rf but increases it at the other parts. The detector network needs accurate conv5_3 features throughout the Rf in order to generate good predictions.

DIRECT again performs the worst. In particular, the performance drops significantly at $\theta{=}18°$, showing that it is sensitive to the distortion. In contrast, INTERP performs better near the pole because the samples are denser on the unit sphere. In fact, INTERP should converge to EXACT at the pole. PERSPECTIVE outperforms INTERP near the equator but is worse in other regions. Note that $\theta{\in}\{18°, 36°\}$ falls on the top face, and $\theta{=}54°$ is near the border of the face. The result suggests that PERSPECTIVE is still sensitive to the polar angle, and it performs the best when the object is near the center of the faces where the perspective distortion is small.

Fig. 7b shows the performance of the object proposal network for two scales (see Supp. for more). Interestingly, the result is different from the detector network. OPTSPHCONV still performs almost the same EXACT, and SPHCONV-PRE performs better than baselines. However, DIRECT now outperforms other baselines, suggesting that the proposal network is not as sensitive as the detector network to the distortion introduced by equirectangular projection. The performance of the methods is similar when the object is larger (right plot), even though the output error is significantly different. The only exception is PERSPECTIVE, which performs poorly for $\theta{\in}\{54°, 72°, 90°\}$ regardless of the object scale. It again suggests that objectness is sensitive to the perspective image being sampled.

Fig. 8 shows examples of objects successfully detected by our approach in spite of severe distortions. See Supp. for more examples.

## 5    Conclusion

We propose to learn spherical convolutions for 360° images. Our solution entails a new form of distillation across camera projection models. Compared to current practices for feature extraction on 360° images/video, spherical convolution benefits efficiency by avoiding performing multiple perspective projections, and it benefits accuracy by adapting kernels to the distortions in equirectangular projection. Results on two datasets demonstrate how it successfully transfers state-of-the-art vision models from the realm of limited FOV 2D imagery into the realm of omnidirectional data.

Future work will explore SPHCONV in the context of other dense prediction problems like segmentation, as well as the impact of different projection models within our basic framework.

## Footnotes

[1]e.g., with equirectangular projection, where latitudes are mapped to horizontal lines of uniform spacing

[2]e.g., $N_p$ could be AlexNet [25] or VGG [33] pre-trained for a large-scale recognition task.

[3]`https://github.com/rbgirshick/py-faster-rcnn`

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
