[Supplementary Material]

# Learning Spherical Convolution
# for Fast Features from 360° Imagery

**Yu-Chuan Su**        **Kristen Grauman**
The University of Texas at Austin

In this file we provide additional details to supplement the main paper submission. In particular, this document contains:

1. Figure illustration of the spherical convolution network structure
2. Implementation details, in particular the learning process
3. Data preparation process of each dataset
4. Complete experiment results
5. Additional object detection result on Pascal, including both success and failure cases
6. Complete visualization of the AlexNet conv1 kernel in spherical convolution

## 1 Spherical Convolution Network Structure

Fig. 1 shows how the proposed spherical convolutional network differs from an ordinary convolutional neural network (CNN). In a CNN, each kernel convolves over the entire 2D map to generate a 2D output. Alternatively, it can be considered as a neural network with a tied weight constraint, where the weights are shared across all rows and columns. In contrast, spherical convolution only ties the weights along each row. It learns a kernel for each row, and the kernel only convolves along the row to generate 1D output. Also, the kernel size may differ at different rows and layers, and it expands near the top and bottom of the image.

## 2 Additional Implementation Details

We train the network using ADAM [1]. For pre-training, we use the batch size of 256 and initialize the learning rate to 0.01. For layers without batch normalization, we train the kernel for 16,000 iterations and decrease the learning rate by 10 every 4,000 iterations. For layers with batch normalization, we train for 4,000 iterations and decrease the learning rate every 1,000 iterations. For fine-tuning, we first fine-tune the network on conv3_3 for 12,000 iterations with batch size of 1. The learning rate is set to 1e-5 and is divided by 10 after 6,000 iterations. We then fine-tune the network on conv5_3 for 2,048 iterations. The learning rate is initialized to 1e-4 and is divided by 10 after 1,024 iterations. We do not insert batch normalization in conv1_2 to conv3_3 because we empirically find that it increases the training error.

## 3 Data Preparation

This section provides more details about the dataset splits and sampling procedures.

**Pano2Vid**    For the **Pano2Vid** dataset, we discard videos with resolution $W \neq 2H$ and sample frames at 0.05fps. We use "Mountain Climbing" for testing because it contains the smallest number of frames. Note that the training data contains no instances of "Mountain Climbing", such that our network is forced to generalize across semantic content. We sample at a low frame rate in order to reduce temporal redundancy in both training and testing splits. For kernel-wise pre-training and

Figure 1: Spherical convolution illustration. The kernel weights at different rows of the image are untied, and each kernel convolves over one row to generate 1D output. The kernel size also differs at different rows and layers.

testing, we sample the output on 40 pixels per row uniformly to reduce spatial redundancy. Our preliminary experiments show that a denser sample for training does not improve the performance.

**PASCAL VOC 2007** As discussed in the main paper, we transform the 2D PASCAL images into equirectangular projected 360° data in order to test object detection in omnidirectional data while still being able to rely on an existing ground truthed dataset. For each bounding box, we resize the image so the short side of the bounding box matches the target scale. The image is backprojected to the unit sphere using $\mathcal{P}^{-1}$, where the center of the bounding box lies on $\hat{n}$. The unit sphere is unwrapped into equirectangular projection as the test data. We resize the bounding box to three target scales $\{112, 224, 336\}$ corresponding to $\{0.5R, 1.0R, 1.5R\}$, where $R$ is the Rf of $N_p$. Each bounding box is projected to 5 tangent planes with $\phi = 180°$ and $\theta \in \{36°, 72°, 108°, 144°, 180°\}$. By sampling the boxes across a range of scales and tangent plane angles, we systematically test the approach in these varying conditions.

## 4   Complete Experimental Results

This section contains additional experimental results that do not fit in the main paper.

Figure 2: Network output error.

Fig. 2 shows the error of each meta layer in the VGG architecture. This is the complete version of Fig. 4a in the main paper. It becomes more clear to what extent the error of SPHCONV increases as we go deeper in the network as well as how the error of INTERP decreases.

Figure 3: Proposal network accuracy (IoU).

Fig. 3 shows the proposal network accuracy for all three object scales. This is the complete version of Fig. 6b in the main paper. The performance of all methods improves at larger object scales, but PERSPECTIVE still performs poorly near the equator.

## 5   Additional Object Detection Examples

Figures 4, 5 and 6 show example detection results for SPHCONV-PRE on the 360° version of PASCAL VOC 2007. Note that the large black areas are undefined pixels; they exist because the original PASCAL test images are not 360° data, and the content occupies only a portion of the viewing sphere.

Fig. 7 shows examples where the proposal network generate a tight bounding box while the detector network fails to predict the correct object category. While the distortion is not as severe as some of the success cases, it makes the confusing cases more difficult. Fig. 8 shows examples where the proposal network fails to generate tight bounding box. The bounding box is the one with the best intersection over union (IoU), which is less than 0.5 in both examples.

Figure 4: Object detection results on PASCAL VOC 2007 test images transformed to equirectangular projected inputs at different polar angles $\theta$. Black areas indicate regions outside of the narrow field of view (FOV) PASCAL images, i.e., undefined pixels. The polar angle $\theta = 18°$, $36°$, $54°$, $72°$ from top to bottom. Our approach successfully learns to translate a 2D object detector trained on perspective images to $360°$ inputs.

Figure 5: Object detection results on PASCAL VOC 2007 test images transformed to equirectangular projected inputs at $\theta = 36°$.

Figure 6: Object detection results on PASCAL VOC 2007 test images transformed to equirectangular projected inputs at $\theta = 18°$.

Figure 7: Failure cases of the detector network.

Figure 8: Failure cases of the proposal network.

# 6   Visualizing Kernels in Spherical Convolution

Fig. 9 shows the target kernels in the AlexNet [2] model and the corresponding kernels learned by our approach at different polar angles $\theta \in \{9°, 18°, 36°, 72°\}$. This is the complete list for Fig. 5 in the main paper. Here we see how each kernel stretches according to the polar angle, and it is clear that some of the kernels in spherical convolution have larger weights than the original kernels. As discussed in the main paper, these examples are for visualization only. As we show, the first layer is amenable to an analytic solution, and only layers $l > 1$ are learned by our method.

Figure 9: Learned conv1 kernels in AlexNet (full). Each square patch is an AlexNet kernel in perpsective projection. The four rectangular kernels beside it are the kernels learned in our network to achieve the same features when applied to an equirectangular projection of the $360°$ viewing sphere.