[Reviews · NeurIPS 2017]

Reviewer 1



This paper describes a method to transform networks learned on perspective images to take spherical images as input. This is an important problem as fisheye and 360-degree sensors become more and more ubiquitous but training data is relatively scarce. The method first transforms the network architecture to adapt the filter sizes and pooling operations to convolutions on a equirectangular representation/projection. Next the filters are learned to match the feature responses of the original network when considering the projections to the tangent plane of the respective feature response. The filters are pre-learned layer-by-layer and fine-tuned to output features as similar as possible to the original network projected to the tangent planes. Detection experiments on Pano2Vid and PASCAL demonstrate that the technique performs slightly below the optimal performance using per-pixel tangent projections (however significantly faster) while outperforming several baselines, including cube map projections. The paper is well written and largely easy to read. While simple and sometimes quite ad hoc (Sec 3.2), I like the idea and am not aware of any closely related work (though I am not an expert in that domain). The technical novelty is quite limited though as it largely boils down to retraining a newly structured network on equirectangular images based on a pre-trained model. I think the experiments still warrant publication (though I would have believed that a pure computer vision venue would be more appropriate than NIPS due to the specific application), so overall I am slightly on the positive side. Detailed comments (in order of the paper): Introduction: The upper bound the method can achieved is defined by running a CNN on all possible tangent projections. This seems fair, but does not deal with very large and close objects which would be quite distorted on the tangent plane (and occur frequently in 360 degree videos / GoPro Videos / etc). I don't think this is a weakness but maybe a comment on this might be good. Related work: I am missing a section discussing related work on CNNs operating on distorted inputs and invariance of CNNs against geometric transformations, eg., spatial transformer networks, warped convolutions, and other works which consider invariance. l. 117: W -> W x W pixels Eq. 2: I find ":=" odd here, maybe just "=" or approximate? Also, it is a bit sloppy to have x/y on the lhs and \phi/\psi on the rhs without formalizing their connection in that context (it is only textual before). Maybe ok though. After this equation there is some text describing the overall approach, but I couldn't fully grasp it at this point. I would add more information here, for example that the goal is to build a network on the equirectangular image and match the outputs after each convolution to the corresponding outputs of the CNN computed on the tangent plane. Fig. 2b: It is unclear to me how such a distortion pattern (top image) can arise. Why are the boundaries so non-smooth (double curved left image boundary for instance). I don't see how this could arise. Fig. 2 caption: It would be good to say "each row of the equirectangular image (ie, \phi)" and in (b) what is inversely projected. l. 131: It is never clearly described how these true outputs are computed from exhaustive planar reprojections. Depending on the receptive field size, is a separate network run on each possible tangent plane? More details would be appreciated. Sec. 3.2: I found the whole description quite ad-hoc and heuristic. I don't have a concrete comment for improvement here though except that it should be mentioned early on that this can only lead to an approximate result when minimizing the loss in the end. l. 171: It is unclear here if the loss is over all feature maps (must be) and over all layers (probably not?). Also, how are the weights initialized? This seems never mentioned. l. 182: It is unclear why and how max-pooling is replaced by dilated convolutions. One operation has parameters while the others has not. l. 191: It is not Eq. 2 that is used but a loss that considers the difference between the lhs and the rhs of that equation I assume? Why not stating this explicitly or better reuse the equation from the beginning of Sec 3.3.? l. 209: What happens to the regions with no content? Are they excluded or 0-padded? l. 222: Why is detection performance evaluated as classification accuracy. Shouldn't this be average precision or alike? Experiments: Some more qualitative results would be nice to shed some light. Maybe some space can be made by squeezing the text partially.

Reviewer 2



Summary ------- The paper proposes an approach for making existing CNNs that have been trained on standard, 'perspective' imagery, available for being applied to 360-degree, spherical imagery. The proposed approach is phrased as a regression problem that mimicks the convolutional response of a 'target network' (such as VGG [25] etc.) on exhaustive tangent plane projections of a spherical image by responses of a 'spherical convolution network' that is applied to a single equirectangular projection. The result is a method that produces convolutions on spherical images that are both accurate (since they resemble the responses on all possible tangent planes) and fast to compute (actual tangent planes are not constructed, but replaced by a single equi-rectangular projection). Apart from the basic idea of formulating the problem of applying existing networks to spherical imagery, the paper proposes a specific NN architecture and a number of technical modifications to traditional NNs that are claimed to be crucial for making the method work: separate convolution kernels are learned for different output rows, kernel shape is relaxed to being rectangular (not square), and pooling layer size is adapted to pixel size. In addition, a training procedure is suggested that pre-trains individual kernels independently before jointly fine-tuning the entire network. Experiments are conducted on existing datasets Pano2Vid [27] and PASCAL VOC, based on existing network Faster-RCNN [22]. Several variants of the proposed method (with and without kernel-wise pre-training) and baselines are compared both w.r.t. accuracy in convolution results (RMSE) and final object class detection performance. To that end, PASCAL VOC images are artificially transformed into spherical images. The proposed method is demonstrated to perform better than the baselines and almost on par with exhaustive tangent plane projections, at a lower computational cost. Novelty and significance ------------------------ To my knowledge, the proposed method is novel in that is the first to propose a learning-based framework for making existing, trained CNNs available for application to spherical images and also in the particular technical implementation. Given the demonstrated performance benefits over the equirectangular projection on one and the computational savings compared to the exhaustive computation of tangent planes on the other hand, it has the potential to advance or at least inspire the next standard method for processing spherical images via CNNs. Technical correctness --------------------- The proposed method seems plausible and technically correct. Experimental evaluation ----------------------- The given experimental evaluation is extensive in terms of examined aspects (baselines, method variations, tasks and applied measures) and convincingly demonstrates both superior performance of the proposed method and computational efficiency in comparison to existing approaches. Specifically, the experiments verfiy both accurate recreation of convolutional responses and benefit for an object detection task. Presentation ------------ The presentation is excellent; the paper follows a clear line of argumentation and is well written.

Reviewer 3



The paper presents an approach to feature extraction on 360-degree images. The approach is to learn a projection operator into R^2, where standard ConvNets take over. The results show a mild accuracy gain due to the learned operator. Overall this is a competently executed and well-written paper. The work strikes me as a bit incremental, but it will be of interest to some people and will not be out of place at NIPS.